# Decentralized Sharing and Valuation of Fleet Robotic Data

**Yuchong Geng**
Cornell University
yg534@cornell.edu

**Dongyue Zhang**
Cornell University
dz356@cornell.edu

**Po-han Li**
The University of Texas at Austin
pohanli@utexas.edu

**Oguzhan Akcin**
The University of Texas at Austin
oguzhanakcin@utexas.edu

**Ao Tang**
Cornell University
atang@cornell.edu

**Sandeep P. Chinchali** *
The University of Texas at Austin
sandeepc@utexas.edu

**Abstract:** We propose a decentralized learning framework for robots to trade, price, and discover valuable machine learning (ML) training data. Today's robotic fleets, such as self-driving vehicles, can gather terabytes of rich video and LIDAR data [1] in diverse, geo-distributed environments. Often, robots in one city or home might observe training data that is commonplace for them but is actually a valuable, out-of-distribution (OoD) dataset to train robust ML models at robots elsewhere. However, simply sharing all this diverse data in cloud databases is infeasible due to limits on privacy and network bandwidth. Inspired by decentralized file-sharing protocols like BitTorrent, we propose a novel system where each robot is provisioned with a learnable privacy filter and sharing model. Importantly, this sharing model attempts to predict and prioritize which sensory percepts are of high value to *other* robotic peers using a decentralized voting and feedback mechanism. Our scheme naturally raises timely questions on data privacy and valuation as companies start to deploy robots in our homes, hospitals, and roads.

**Keywords:** Decentralized Fleet Learning, Swarm Robotics, Data Valuation

## 1 Introduction

Imagine a fleet of robots that are deployed at diverse hospitals and nursing homes across the world. These robots should seamlessly interact with nurses and patients that speak different languages, have diverse preferences on human-robot interaction, and rapidly adapt to new hospital layouts and visual appearances of medical equipment. To do so, these robots can continually re-train their perception, natural language processing, and control policies using hundreds of gigabytes of rich video, audio, and LIDAR sensory streams they observe locally. Moreover, these robots should ideally share their rare, valuable training data with their peers in diverse locations in order to train more robust ML models. However, how should we incentivize heterogeneous robots (often owned by different companies) to share valuable training data while protecting human privacy and proprietary ML models?

Designing private, efficient data-sharing protocols is of prime importance today. Already, the Toyota Research Institute is advocating large-scale fleet learning in homes [2], Diligent Robotics is deploying robot swarms to assist nurses with fighting COVID-19 [3], and platforms like SafetyPool allow multiple self-driving car companies to crowd-source hazardous driving scenarios [4].

Despite this potential, our key observation is that today's mechanisms for data sharing are largely insufficient since they are *centralized and data-agnostic*. In other words, we typically share *all* data with trusted cloud servers regardless of whether it is useful to other robots, a redundant concept in our training dataset, or private. As such, today's mechanisms can lead to privacy concerns and waste valuable network bandwidth, cloud storage, and cloud training resources. In this paper, we argue that individual robots should have sovereignty over their data and sharing should be **selective, private, and decentralized**.

---

*Corresponding author

Blue Sky Papers, 5th Conference on Robot Learning (CoRL 2021), London, UK.

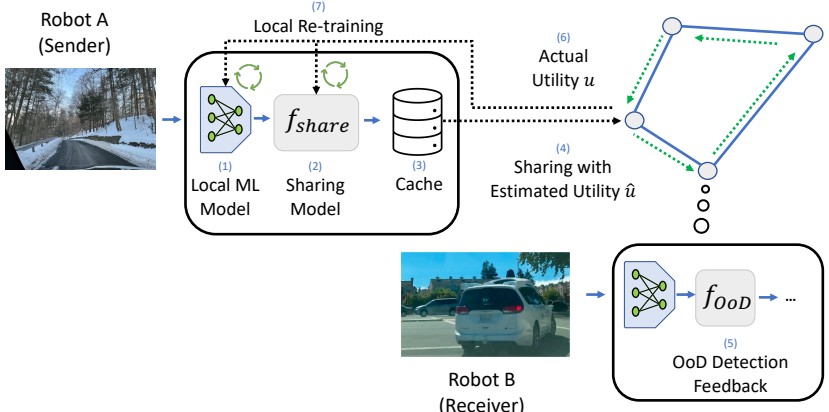

Figure 1: We collected dashcam videos from 3 diverse rural and metropolitan areas thousands of miles apart. For example, a car in a rural snowy area should have a specialized vision model for snowy terrain. While it might find a new image of snow commonplace, that very same image could be OoD and valuable to a peer thousands of miles away in a sunny environment to build a more robust model. As such, our system would *proactively* share this data using a global utility (GU) model (steps 1-3) and receive feedback from networked peers (steps 4-7).

**The State-of-the-Art:** Historically, robotic fleet learning has been addressed by related work in cloud robotics [5, 6, 7, 8], active learning [9, 10, 11, 12], and crowd-sourced platforms such as RoboTurk [13] and RoboEarth [14]. However, these mechanisms rely on trusted cloud servers to aggregate data and do not address privacy or how to fairly assess the contribution of each training example to overall model accuracy. Moreover, today's approaches to privacy in ML, such as differential privacy (DP) [15, 16] and federated learning (FL) [17, 18], do not readily apply to robots that operate amongst humans. For example, DP approaches would add excessive noise to anonymize high-dimensional robotic visual data, which severely limits its utility for training ML models. Likewise, FL allows mobile devices to train ML models on local private data and simply share gradient updates with a central cloud server. However, FL does not apply when a robot sees a novel image it cannot classify locally and must share this image with a peer to receive a ground-truth label. Finally, our proposal is inspired by classical work on consensus in multi-agent systems [19], which pre-dates our focus on trading data for ML models.

**Contributions:** Given the shortcomings of today's methods, our principal contribution is to propose a communication-efficient, private, and proactive data sharing system. By proactive, we mean that data sharing can potentially allow robots to improve their models on otherwise unseen data and thus avoid failure before they actually encounter such scenarios. We now describe our proposal in detail.

## 2 System Architecture and Preliminary Results

Figure 1 depicts our modular system architecture. Each robot in a peer-to-peer network acts both as a sender as well as a receiver of training data. For clarity, Figure 1 illustrates a simple sharing scenario where a sender robot (left) shares presumably valuable data to receivers (right), which in turn provide feedback to the sender to adjust the sharing process. Crucially, to limit network bandwidth, training, and data annotation costs, each robot is limited to share only 1-5% of local data with its peers. Data sharing and model re-training occur in periodic rounds, such as the end of a day. While our architecture is broadly applicable to many ML models, we consider a scenario in the self-driving domain. Here, a robot in a rural snowy area shares data with a vehicle in a sunny city to build a robust ML model across multiple weather conditions. We now describe the information flow in one round of data sharing by referencing the numbered blocks in Figure 1.

**1. Local ML Model.** Each robot starts with a generic ML model pre-trained on anonymized public datasets like ImageNet. However, each robot can specialize this model based on its own local data with assistance of trusted, local data annotators. In the context of robotic perception, the specialized

vision model maps a new image observation $X$ into a predicted class or object detection $\hat{y}$ and embedding vector $e$.

**2. Global Utility (GU) Sharing Model.** The crux of our approach is to provision each robot with a global utility (GU) model which learns the value (e.g. utility) of a new image $X$ to other robotic peers. One example of a utility score is the proportion of other robots in a fleet that consider image $X$ to be out-of-distribution with respect to their current vision model, which could indicate a valuable training scenario. To estimate the utility of a specific image $X$, the GU model naturally takes in the vision model's embedding $e$ and predictions $\hat{y}$. Crucially, the GU model has learnable parameters $\theta_{\text{share}}$ that allow it to predict a utility score $\hat{u}$ per image. While the utility might be uniform initially, our next steps show how to estimate the true utility for each image using prioritized *network feedback*.

**3-4. Cache, Privacy Filter, and Network Transmission.** After images are ranked by their utility, the next step is to prioritize them for sharing while respecting bandwidth limits. As such, each robot only caches the top $1 - 10\%$ of images with the highest utilities $\hat{u}$ in its local storage. Then, these cached images can be blurred or altogether dropped to restrict sensitive human faces or other private features. Finally, each sender can now relay the cached and anonymized images to other peers.

**5-6. Network Feedback Based on Out-of-Distribution Detection.** The next key step is to have other robots evaluate the utility of shared data in order to eventually tune the GU model. Each shared image $X$ has a default *true* utility counter $u$, initialized to zero, to indicate how many robots found image $X$ interesting. Each robot then uses an OoD detector [20, 21, 22, 23, 24, 25] to determine if shared image $X$ displays novel features far from its local training distribution. If so, this robot increments the utility $u$ before relaying image $X$ and the current utility to new robots. Finally, the utility $u$ is normalized to represent the fraction of peers that found the data interesting. This feedback is then relayed back to the sender robot to re-train its GU model.

**7. Closing the Loop: Re-training Sharing and Local ML Models.** Crucially, each sender robot has now been relayed a dataset of its original shared images $X$, their embeddings and predictions $e, \hat{y}$, estimated utility $\hat{u}$, and true utility $u$ from peers. As such, we can re-train the GU model to update parameters $\theta'_{\text{share}}$ to improve the sharing process. Moreover, each robot can re-train its vision model with new data it received with possible help from trusted local data annotators. Ideally, this process will converge when each local vision model is more robust to previously OoD data.

**Preliminary Evaluation** We now describe a simple toy illustrative example of data sharing. We consider a scenario with three networked 'robots', each of which has a pre-trained vision model to detect handwritten digits from the benchmark MNIST dataset. Crucially, to simulate the presence of rare, valuable training data, one robot also observes data from the KMNIST [26] variant of MNIST while another observes data from FashionMNIST [27], which both exhibit distinct visual properties from standard MNIST. Our goal is to train a GU model that learns to share KMNIST and FashionMNIST data between appropriate robots so that all robots have access to all classes of training data. Our evaluation metric is the final accuracy of each robot's vision model on all three variants of MNIST to demonstrate robust performance on data each robot was not initially exposed to.

We implemented the data sharing protocol illustrated in Figure 1. First, each robot passes a new image into its local vision model, whose embedding and predictions are passed to the GU sharing model. Since the GU sharing model is initially un-trained, it first randomly caches images to share with other peers. Importantly, each robot can only share 10% of its training data per round due to network bandwidth limits. As data is shared, each peer provides a utility score $u$ based on whether a shared image was valuable (e.g. OoD). Concretely, this OoD detector uses recently-developed Bayesian uncertainty estimates to provide a scalar OoD signal to indicate if a new image is from the local vision model's training distribution. After the utility score for shared images is relayed back to the sender, the sender re-trains its GU sharing model to prioritize relevant content.

**Results:** Figure 2 illustrates strong experimental results for our toy example of data sharing. We compare our protocol based on a GU sharing model (green) with upper and lower bounds of oracle and random sampling in blue and red respectively. In particular, the un-realizable oracle approach has a perfect GU model that shares OoD data with perfect accuracy. In stark contrast, the random policy is a lower-bound that wastes precious network bandwidth by randomly sharing images as opposed to prioritizing them based on network feedback. Our key evaluation metric is the *overall multi-class* accuracy (y-axis) across data sharing and training rounds on the x-axis.

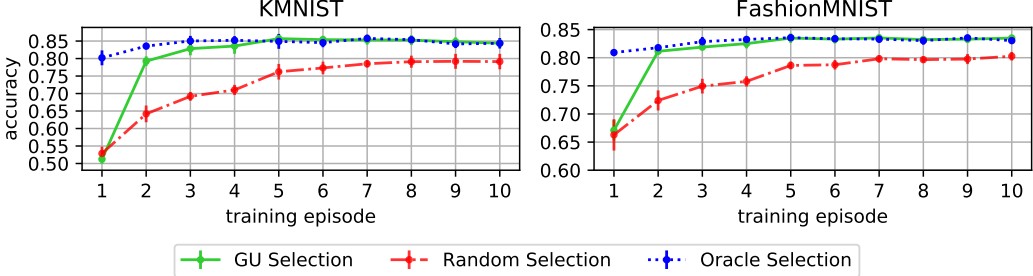

Figure 2: **Toy Experiment Results:** Initially, a robot's local vision model does not have access to KMNIST and FashionMNIST data and hence performs poorly on these OoD concepts in round 0. Clearly, our learned sharing model (green) allows a robot to quickly accrue valuable training data from its peers, which allows the local vision model to efficiently be re-trained to match the upper-bound accuracy of an oracle data collection strategy (blue).

Clearly, our decentralized sharing protocol (green) significantly improves each robot's vision model by sharing previously OoD training data with peers that need it most. Moreover, our scheme is efficient and adaptive – the GU model learns to correctly cache 88.18 % of rare KMNIST data and 88.96% of rare FashionMNIST data, which is significantly higher than random sharing.

## 3    Future Research Questions on Data Valuation and Privacy

We now describe open research questions that cross-cut ML, privacy research, and cloud robotics.

**Game-Theoretic Approaches to Data Valuation** In order to incentivize data sharing, each robot should be paid a fair price for the data it shares. To do so, we can adapt the recently-proposed Data Shapley [28] metric, which is inspired by cooperative game theory. Specifically, this metric considers each training example to be a player in a cooperative game, which is to collectively train a large ML model. The payout or value to each player (i.e., data point) is calculated based on the marginal accuracy gain when that specific data example is used for training as opposed to when it is omitted, averaged over all possible subsets of training data where a specific example can appear. While promising, today's Data Shapley metric only scales to small, centralized datasets and a homogenous ML model. However, our network feedback scheme is a natural approach to value data based on the collective score assigned by peer robots' OoD detectors. Future research can reconcile our data valuation metric and extend Data Shapley to decentralized, heterogenous ML models.

**Privacy-Utility Trade-offs in Robotics** Today's federated learning [18] approaches make a strong assumption that ground-truth labels are present on mobile clients, which allows them to train models locally on private data and simply share anonymized gradient updates. However, such a restrictive assumption is infeasible in robotics - if a robot sees a novel scene it cannot classify locally, it must necessarily share the raw training data to ask for a label. Future research should investigate how to blur sensitive information before sharing, such as in Google Street View [29], but for a broad set of private classes. Moreover, we believe (but must certify) that our system guards against malicious actors that falsify their utility for data, since this will cause them to see less relevant data over time.

**Formal Convergence Analysis** Future work should establish whether our data sharing protocol converges to a steady state where rare training data is evenly distributed across robotic peers. Our preliminary theoretical analysis estimates the number of rare images cached by each robot based on the precision and recall of the GU model and OoD detectors for each round of data sharing.

**A Data-Addressable Network?** Inspired by internet protocols, we propose a Data Discovery Service where robots can quickly identify new peers joining the network that are likely to have valuable training data for a task of interest. Likewise, future research should explore whether OoD detectors and our voting scheme can scalably run on commodity networks and resource-constrained robots.

Overall, robots are poised to leave the confines of the factory floor to enter our homes, hospitals, and airspace. Given the diversity of data owners and interactions with humans, it is increasingly important to consider privacy and data valuation as key desiderata in robotic fleet learning.

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
