# OpenReview forum: "Decentralized Sharing and Valuation of Fleet Robotic Data"
_robot-learning.org/CoRL/2021/Conference/Blue_Sky — CoRL 2021, Blue Sky_

### Official Review · Reviewer_F5uz · 2021-08-12

**Novelty:** Very Good
**Impact:** 2
**Clarity Of Presentation:** Good

**Recommendation:**

Weak Accept: I recommend accepting the paper, but will not argue for my recommendation if the majority of other reviewers have a different opinion.

**Summary:**

The authors make the case for a decentralized data sharing system that in addition to preserving privacy of the data, enables to value and prioritize data for other robots. The paper also presents initial results of an implementation of this idea and provides an outlook on future work.

**Summary Of Recommendation:**

This paper presents an interesting problem that not many people in the robot learning community are thinking about. I am not an expert on the topic of data privacy and sharing but the ideas presented in the manuscript make sense to me. Please see more detailed points below:

- There are many companies open-sourcing their data now, including data useful for business applications such as autonomous driving. I would be curious to hear the authors' thoughts on the alternative where one pre-trains their model on a large, publicly-available dataset. I would think that in this case the data sharing aspects are not as important, since the algorithms would require very little data to adjust to new situations.
- It is unclear to me why in Fig. 2 the random strategy gets better with more training episodes. Please provide additional analysis of this figure.

---

### Official Review · Reviewer_9fPg · 2021-08-26

**Novelty:** Good
**Impact:** 3
**Clarity Of Presentation:** Very Good

**Recommendation:**

Weak Accept: I recommend accepting the paper, but will not argue for my recommendation if the majority of other reviewers have a different opinion.

**Summary:**

The paper addresses the challenge of how to combine and utilize data from several different robotic sources efficiently. A requirement in this proposal is that the robots cannot directly share all collected data with each other but need to filter the data before sharing. The paper argues that sharing should be selective, private, and decentralized.

The paper proposes an approach where robots learn a utility model of data so that the robots only transmit and store information that is valuable to other robots. The paper assumes that out-of-distribution data is valuable to robots since they can learn a more robust policy from that data. Robots may choose to transmit only data that is not private to them or otherwise secret from other robots.

In a proof-of-concept experiment evaluation, robots have access to different image databases. In the evaluation, robots learn which data is valuable to other robots and the proposed approach outperforms baselines.

**Summary Of Recommendation:**

The motivation and reasoning for the decentralized data distribution framework among robot peers makes sense. Taking value of the data and privacy into account and also thus using a decentralized model can be valuable.

The proof-of-concept experimental comparison of baselines and the proposed approach is valuable.

I am not fully convinced of the chosen approach for distributing the data, discussed in more detail below. I recommend to discuss this in some way in the next revision of the paper.

Since agents transmit out-of-distribution data to other agents, an assumption behind the proposed framework is that the goal is to optimize the behavior of all agents in all environments. Is this a beneficial / realistic assumption? Would it not be more efficient to allow robots to learn to cope in only those environments that they encounter as well as possible? In practice, this would mean peers could request the data they need when needed. For example, when an agent sees an out-of-distribution data sample in its current environment, it would send a request to get more data that is similar to this OoD sample. This would also remove the need for learning a utility model, reduce network traffic, and allow robots to advertise the price they are willing to pay for data.

Note: I would rephrase my Impact score as: "The work is incremental but may have much impact".

---

### Decision · Program_Chairs · 2021-10-01

**Decision:**

Accept

**Comment:**

The paper discussed an idea of data-sharing in the multi-robot fleets and the importance of developing methods that make the data exchange privacy-enabled, and also filter the most relevant that to be shared with the peers. The reviewers agree that the paper will well-written, and provides a fresh new and not studied idea in robot learning. The investigation in this area can have a big impact on the field of the robot learning.